# Cold Atmospheric Plasma Medicine: Applications, Challenges, and Opportunities for Predictive Control

Ali Kazemi [1] , McKayla J. Nicol [2,3,4], Sven G. Bilén [5] , Girish S. Kirimanjeswara [2,6,7,*] and Sean D. Knecht [5,*]

1   Biomedical Engineering Graduate Program, The Pennsylvania State University,
    University Park, PA 16802, USA; ajk5858@psu.edu
2   Department of Veterinary and Biomedical Sciences, The Pennsylvania State University,
    University Park, PA 16802, USA
3   Pathobiology Graduate Program, The Pennsylvania State University, University Park, PA 16802, USA
4   Clinical and Translational Sciences Graduate Program, The Pennsylvania State University,
    University Park, PA 16802, USA
5   School of Engineering Design and Innovation, The Pennsylvania State University,
    University Park, PA 16802, USA; sbilen@psu.edu
6   The Center for Molecular Immunology and Infectious Disease, The Pennsylvania State University,
    University Park, PA 16802, USA
7   The Center for Infectious Disease Dynamics, The Pennsylvania State University,
    University Park, PA 16802, USA
*   Correspondence: gsk125@psu.edu (G.S.K.); sdk149@psu.edu (S.D.K.)

**Abstract:** Plasma medicine is an emerging field that applies the science and engineering of physical plasma to biomedical applications. Low-temperature plasma, also known as cold plasma, is generated via the ionization of atoms in a gas, generally via exposure to strong electric fields, and consists of ions, free radicals, and molecules at varying energy states. Plasmas generated at low temperatures (approximately room temperature) have been used for applications in dermatology, oncology, and anti-microbial strategies. Despite current and ongoing clinical use, the exact mechanisms of action and the full range of effects of cold plasma treatment on cells are only just beginning to be understood. Direct and indirect effects of plasma on immune cells have the potential to be utilized for various applications such as immunomodulation, anti-infective therapies, and regulating inflammation. In this review, we combine diverse expertise in the fields of plasma chemistry, device design, and immunobiology to cover the history and current state of plasma medicine, basic plasma chemistry and their implications, the effects of cold atmospheric plasma on host cells with their potential immunological consequences, future directions, and the outlook and recommendations for plasma medicine.

**Keywords:** cold atmospheric plasma; CAP; low-temperature plasma; LTP; ROS; RNS; RONS; plasma medicine; biological effects; dose; cold plasma

## 1. Background

Physical plasma is generated via the ionization of particles within a gas, generally via exposure to extremely high temperatures, high-energy electrons or photons, or strong electric fields [1]. This exposure creates a quasi-neutral fluid consisting of free electrons, ions, free radicals, and atoms/molecules at varying energy states, accompanied by ultraviolet emissions and an electric field. Physical plasma—hereafter referred to simply as plasma in this review, as opposed to blood plasma—is thought to have been first produced purposefully in the laboratory in 1879 by Sir William Crookes; however, the term "plasma" was not introduced until 1928 by chemist Irving Langmuir [2]. Since these early investigations, plasma research has developed into a complex and multifaceted area of study with well-established applications in semiconductor manufacturing [3,4], clean energy [5,6], lighting [7], space propulsion [8], and many others. The focus of this review is

the biomedical applications of cold atmospheric plasmas (CAPs) and the current obstacles in the emerging field of plasma medicine (a literature search was conducted with a focus on publications between 2017 and 2023, with more emphasis on the later years).

CAP is generated when the energy of electrons in the plasma is quite high in comparison to low-energy ions and molecules within the gas, resulting in a non-equilibrium plasma discharge. These low-temperature or non-thermal plasmas can be applied to biological substrates without thermal damage [9]. CAP treatment of biological substrates can elicit a variety of responses ranging from programmed cell death (apoptosis) to proliferation of cells. It is hypothesized that reactive oxygen and nitrogen species (RONS), particularly those generated via secondary reactions during plasma generation, are of the utmost importance in inducing biological activity within various forms of plasma treatments [10]. However, as mentioned, CAP generation is accompanied by electric fields and photon emission, the physiological effects of which cannot be ignored. The variability of CAP, coupled with highly target-specific responses and multi-scale physico–chemical and biological phenomena, has introduced significant complexity in defining a unit dose for plasma therapeutics, which is an important obstacle to the mainstream adoption of CAP technologies within the clinic.

## 1.1. History and Current State of Plasma Medicine

Advances in technology, including compact, solid-state high-voltage power supplies, have enabled consistent generation of these cold, or non-equilibrium, plasmas at atmospheric pressures. The primary classifications for plasma generation that we consider are thermal versus non-thermal and low-pressure versus atmospheric pressure plasmas [10]. In thermal plasma, the electrons, ions, and any neutral species have reached thermal equilibrium with one another, often at high temperatures (thousands of K and greater). However, in non-thermal plasma, the electrons exist at high energies (eV to 10 s of eV), while the more massive ions and neutrals remain close to ambient temperature [11]. Low-pressure plasma indicates that the operating conditions are well below atmospheric pressure, such as those employed in nuclear fusion, semiconductor manufacturing, and space propulsion. The identification and further investigation of non-thermal, atmospheric pressure plasma that can be applied to soft or living surfaces without thermal damage have resulted in the inception and expansion of the field of plasma medicine, beginning approximately two decades ago.

Plasma medicine has since evolved into a highly interdisciplinary field of research investigating opportunities for applying plasma science and engineering to biomedical problems. Clinically, thermal plasma was originally implemented for cauterization and blood coagulation [12], but the potential for utilizing a non-thermal plasma in healthcare has fostered many concepts for tissue treatment without inducing thermal damage. Presently, the most commonly researched applications within the field of plasma medicine are dentistry [13], dermatology [14,15], oncology [16], infection control, and sterilization [17–19]. However, there is further potential for applications of cold plasma treatments in a variety of other areas, including ophthalmological and neurological applications [20–22]. Some applications of plasma medicine extend beyond the realm of tissue or clinical treatments. These applications include surface modifications for implant materials [12], disinfection of equipment or surgical instruments [14], and aerosol decontamination [23,24]. It is important to note, however, that the cold plasma sources that can or should be utilized are distinct depending on the application. Direct plasma treatments, in which the plasma plume is in contact with the target surface for non-living tissue are less restricted in comparison to treatments for live cells or tissue. For example, treatments intended for live cells cannot require low-pressure conditions that might compromise host cell viability. Although significant progress has been made within the field of plasma medicine, there remains a lack of knowledge about in vivo interactions with cold plasma constituents. Specifically, disease prognosis and the full impact of CAP treatments cannot be appropriately assessed

without a more comprehensive understanding of CAP's interactions with both healthy and damaged tissues or cells.

### 1.2. Common Discharge Systems for Cold Atmospheric Pressure Plasma

The most common methods for producing cold plasma under atmospheric pressure conditions include dielectric barrier discharges (DBDs) and atmospheric pressure plasma jets (APPJs) [25]. There are two primary types of DBD configurations shown schematically in Figure 1: volume and surface. A volume DBD plasma is formed in the space separating two electrodes, whereas surface barrier discharge (SBD) plasma appears attached to the exterior of the device. A traditional volume DBD device comprises two synthetic electrodes made of conducting materials, which can include copper, stainless steel, and other conductors. Dielectric coatings for these devices are made from a range of possible materials, including glass, polyimide films, aluminum oxide, various polymers (e.g., Teflon, PVC, etc.), and others (Figure 1B). The size of the treatment volume is limited by power supply capabilities, introducing obstacles in their application for biomedical purposes. The floating-electrode DBD (FE-DBD), shown in Figure 1A, utilizes a dielectric-coated high-voltage electrode that is free to move along the treatment surface. The treatment subject is employed as the counter electrode, and volumetric plasma can then be applied to a larger area [26,27]. An SBD uses a dielectric-coated, high-voltage electrode with the counter electrode attached to the surface of the dielectric. The two electrodes are separated from one another by a dielectric layer, and plasma is generated on the dielectric surface above the high-voltage electrode (Figure 1C). Similar to the FE-DBD, the SBD is free to move and can be used on larger treatment areas. However, for SBDs, the active plasma might not make direct contact with the area of treatment, as the counter electrode is isolated from the designed surface structure [10]. The distance between the device and the treatment surface is important in the SBD configuration. As the device moves farther away from the targeted surface, the treatment becomes less effective. This can likely be explained by the mechanics of particle diffusion in gas and the mean free path of ions. As the target moves away from the plasma source, the time needed for an effective concentration of RONS to travel to the target, as well as the probability of collisions and recombination, increases; at a certain distance (dependent on the intensity of SBD discharge), this will become impossible due to the rapid relaxation of reactive species.

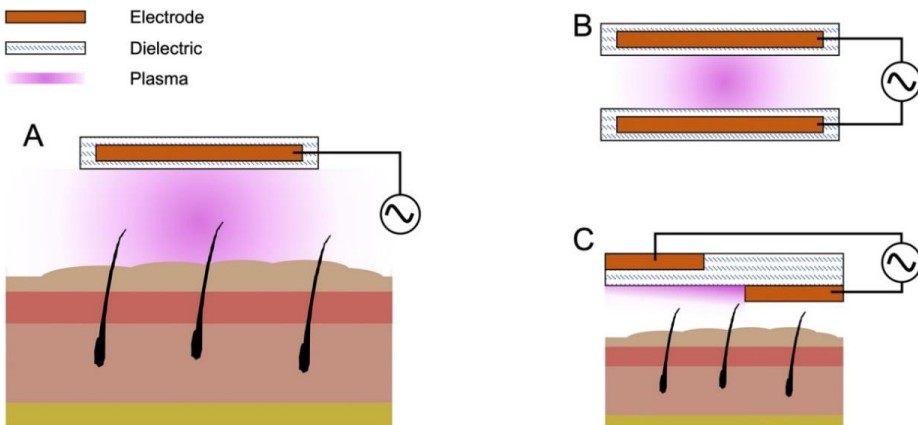

**Figure 1.** Schematic for three dielectric barrier discharge configurations that are most commonly used in biomedical applications. (**A**) Floating-electrode DBD (FE-DBD). Uses a floating high-voltage electrode and employs the treatment target as a counter electrode; (**B**) Volume DBD. Uses a high-voltage electrode and a grounded counter electrode ionizing the volume of gas in between the two; (**C**) SBD. High-voltage and counter electrodes are on the opposite sides of a dielectric barrier, producing discharge on the surface of the SBD, according to the geometry of the electrodes.

APPJs, shown schematically in Figure 2, differ from both volume DBDs and SBDs in that they require the incorporation of carrier gas at a determined flow rate, whereas atmospheric DBDs typically incorporate ambient air as the working gas. Electric fields produced between two synthetic electrodes, with a potential difference on the order of several up to tens of kilovolts, accelerate electrons to high energy, ionizing some fraction of the carrier gas. The plasma is then carried from the device to the area of treatment by means of the carrier gas stream in what is known as a plasma plume. There are two main configurations of APPJs. One configuration incorporates two ring electrodes around the tubing of the apparatus (Figure 2A). In contrast, center-pin jets (Figure 2B) use one ring electrode and a second electrode, which may or may not be encased inside a dielectric, extending down the center of the tube, creating an annular discharge region. Noble gasses are frequently used for APPJ configurations as their atomic nature allows for plasma discharge at relatively lower voltages while maintaining a low temperature. However, pure carrier gases are limited in ROS/RNS production in the plasma plume since the only source of oxygen and nitrogen is the boundary layer of the plume exiting the device. As a result, incorporation of oxygen, nitrogen, or air at small mole fractions in the carrier gas can be beneficial to biomedical applications [9,28].

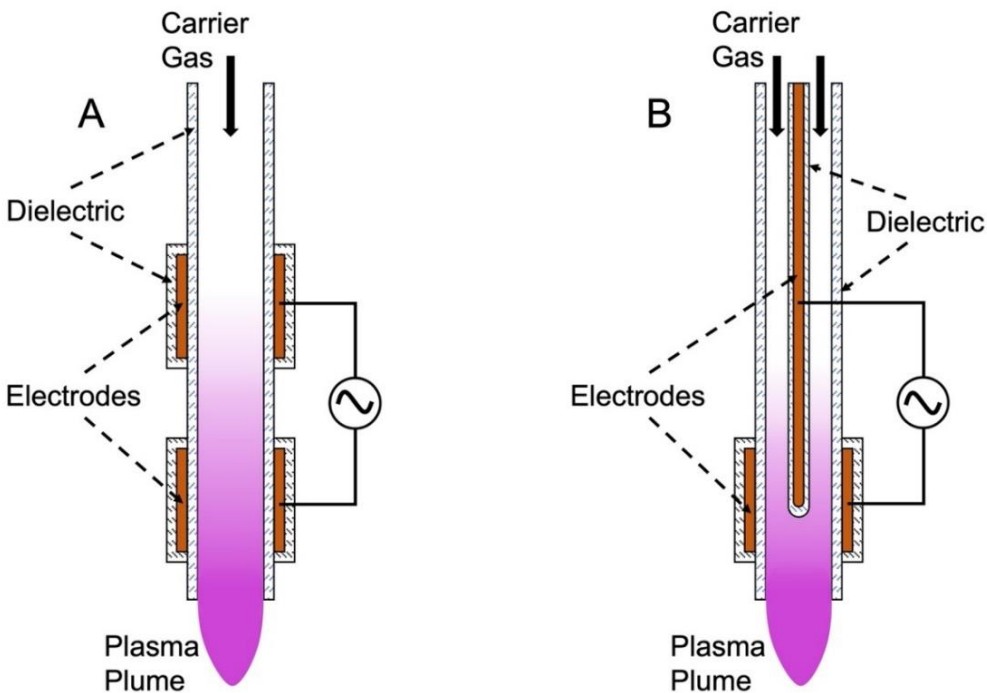

**Figure 2.** Atmospheric pressure plasma jets. (**A**) External ring electrode APPJ; (**B**) center-pin APPJ schematic for two configurations of APPJs that are most commonly used in biomedical applications. Arrows display the direction of carrier gas flow.

Cold plasma sources can be powered in a variety of different ways. Some of the more common plasma power supply regime employed in plasma medicine include low-frequency alternating current (AC) excitation, radiofrequency (RF) excitation, and nanosecond pulsed excitation with other methods such as direct current (DC) and microwave plasmas utilized less commonly.

Presently, there are three plasma devices that have been fully certified for clinical applications. These are the kINPen® MED (Leibniz Institute for Plasma Science and Technology–INP Greifswald and neoplas med GmbH, Greifswald, Germany), PlasmaDerm® VU−2010 (CYNOGY GmbH, Duderstadt, Germany), and Adtec SteriPlas (Adtec Healthcare, Middlesex, UK). The kINPen MED is an APPJ that incorporates an argon gas feed and is mainly targeted towards the treatment of chronic wounds and skin disorders [29]. The

PlasmaDerm uses a DBD configuration that has been investigated for the treatment of chronic leg ulcers [30]. Lastly, the Adtec SteriPlas is another APPJ device that utilizes argon feed gas. This device is mainly used for wound sterilization and treatment of dermatological conditions [31]. These three devices are the only commercially available CAP devices approved for clinical use; however, several more devices are currently under investigation and in approval processes [32]. One example is Canady Helios Cold Plasma (CHCP), which was the first cold plasma device for the treatment of advanced solid tumors to undergo an FDA-approved phase I clinical trial [33]. kINPen MED, SteriPlas, and CHCP operate in the RF range, while the PlasmaDerm is powered by a low-frequency alternating current power supply.

Plasma treatments are also complicated by the requirement of a continuous supply of energy, as it is not possible to store plasma in the gaseous form. To overcome this complication, the use of plasma-activated water or media (PAW or PAM), which are substrates that have retained CAP chemistry after direct exposure to a plasma source, is under investigation [34]. Water, saline solutions, growth media, and hydrogels are some of the substrates that have been demonstrated to be effective media for PAM generation [35]. Various plasma sources can be used for PAM generation, including APPJs, DBDs, microwave plasmas, etc. [36]. PAM treatments have been tested for efficacy in cancer therapy [37], wound treatments [38], and pathogen inactivation [9,39]. PAM hydrogels are especially attractive due to their localized effect and their applicability as a delivery method combining CAP therapy with other pharmaceuticals [37]. Another promising application of PAM is for the treatment of bacterial biofilms. PAM can improve the diffusion of RONS into biofilms and disrupt their structural integrity [39]. Also under investigation are the labile properties of PAM, which are dependent upon storage conditions [40]. PAM is also being investigated for applications within agriculture, food processing, and packaging [41].

*1.3. The Main Effectors of CAP Treatment*

It is thought that the main effector of CAP treatments is RONS that are generated within the plasma discharge [9,11,42]. Figure 3 shows a schematic of how plasma-generated RONS interact with tissue. A plasma source directed at a tissue target creates reactive species and charged particles, including RONS, in the gas phase that then interacts with a surface, which may be liquid or water-filled soft tissue of a patient. The gas-phase RONS, which may include short-lived species such as singlet oxygen, hydroxide, and nitric oxide, react with the much denser liquid/tissue. The resulting secondary reactions cause the conversion of short-lived species into more stable species, such as hydrogen peroxide and nitrite/nitrate. Although these more stable species have a higher likelihood of interacting with treatment targets and provoking a cellular response, the biochemical reactions involving short-lived, plasma-generated species cannot be neglected [43]. However, in situ and even in vitro measurements of these components are often beyond the limits of current diagnostic tools. Several groups have attempted to investigate the penetration depth of CAP treatments in tissue models in hopes of further clarifying some of the biological effects that long- and short-lived species have [44,45].

Endogenous RONS generated by cells within the targeted tissues may also contribute to the treatment effect [42]. It has long been established that ROS/RNS are causative agents of oxidative stress, which is frequently associated with cellular damage and other detrimental effects [46]. It is important to note that these species are also important for normal cell functions and signaling [46]. Additionally, RONS can contribute to healing processes by increasing cell proliferation or inducing death in infected or otherwise damaged cells [46]. A comprehensive knowledge of the correlation between cold plasma input parameters (e.g., intensity of electric fields, frequency, ambient environment, treatment distance, etc.) and plasma chemistry is lacking. For example, an intuitive understanding of plasma generation suggests that increasing the intensity of the applied electric field could lead to a higher concentration of chemical species generated and even enable the generation of species at higher energy states. Increasing the frequency of the applied electric field could increase

the ionization rate while decreasing the power transfer efficiency. Ambient conditions, such as air composition and humidity levels, could play a significant role in RONS generation rate and discharge intensity. Finally, as mentioned in earlier sections, larger treatment distances, depending on the plasma source, could reduce the concentration of RONS at the treatment surface. The establishment of correlations between input parameters, chemistry of discharge, and resulting treatment response of biological targets is an essential precursor to meaningful control of CAP treatment outcomes and the adoption of this technology in medicine. Additionally, there is a lack of understanding of the subsequent systemic effects, side effects, and responses to CAP treatments. Further, in situ investigation is necessary to elucidate such effects and responses.

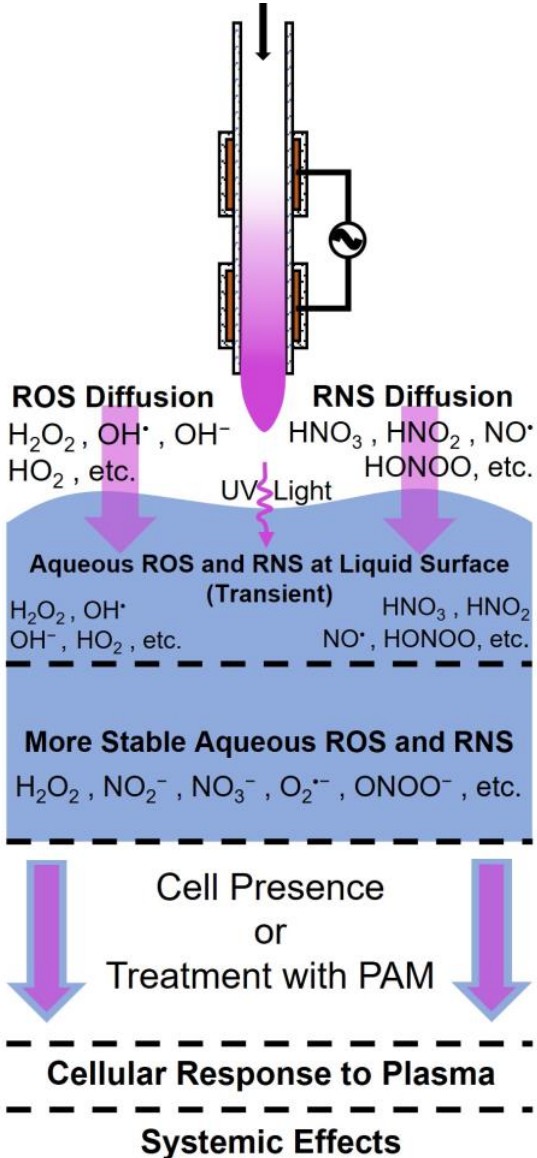

**Figure 3.** CAP-generated reactive species. Plasma-generated ROS and RNS are created in the gas-phase and then interact with a surface, such as water-filled tissue, reacting to produce more stable RONS species, which provoke a cellular response, which may further result in systemic effects in living organisms. Arrows depict diffusion, reaction, or treatment (blue: treatment substrate, violet: plasma constituents).

Although RONS have been identified to play a major role in eliciting the effects of CAP treatments, there are many other characteristics of plasma discharge that should not be overlooked for their potential to play mechanistic roles in treatment outcomes. This includes, but is not limited to, the electric field and UV radiation generated [9]. In fact, it has been speculated that these aspects could be involved in the promotion of electroporation of cell membranes, induction of apoptosis or necrosis, and DNA damage.

### 1.4. A Unified Definition of Plasma Dose Is Essential

The general objective of CAP treatments within the field of plasma medicine is to impact the structural and functional characteristics of cells and tissues of interest. However, therapeutic effects have yet to be fully measured and functionally controlled, especially across different discharge systems. This normalization between treatment types and plasma sources may prove to be intricate as the physiochemical properties of CAPs are dependent on various parameters within each setup [47]. These parameters include the carrier gas or composition of the atmospheric air at the point of plasma generation, electrode configuration, pressure, the power applied to the system, and more [32,48,49]. Adding to the complexity of treatment normalization is the wide range of effects that can be elicited by plasma treatments. Thus far, research has indicated that exposure to cold plasma can both stimulate and inhibit cellular functions [50,51]. Distinction between the two effects will require a correlation between the varying parameter settings of plasma sources and the observed outcomes. These combined factors have shown a hormesis effect of cold plasma treatments, shown conceptually in Figure 4, in which smaller doses of plasma treatment may lead to cell proliferation, whereas larger doses may induce apoptosis or necrosis. The established biphasic response to CAP treatments leads us to the question of how much CAP treatment is too much, and, of course, this is dependent on the desired outcome. For instance, when treating an infected wound, the desirable outcome is the elimination of the pathogen without damaging host cells or interrupting the healing process. To be able to accurately define this limit, meaningful control of the CAP treatment and development of precise, time-resolved measurement techniques with a wide spatial resolution is necessary. Although such technology would not be capable of real-time measurements and, therefore, measurement of plasma dose, its in vitro application could advance our understanding of cold plasmas. Furthermore, current methods of CAP treatment are either spatially non-homogeneous or have very small effective areas and require lateral movement across a larger surface, making normalization of treatment difficult. Adding to the complexity is the fact that different cell types respond differently to identical treatments. Considering that RONS are the primary bioactive component of CAP discharge and adding the major role of redox reactions in maintaining homeostasis, the source of treatment response variability across different physiological systems becomes apparent. Further complexity arises when we account for the nature of plasma discharge, its rapidly changing chemistry, its variability based on a wide range of input parameters, and the limitations of current diagnostic techniques and equipment. The high degree of variability associated with every aspect of plasma medicine, ranging from devices and input parameters to target specific outcomes, emphasizes a vital need for a unified dose definition [52].

Promising results in plasma therapeutics designed for a variety of cancers, pathogens, skin maladies, and dental applications have raised the question of how broad the applicability of CAP within medical sciences is. The following sections focus on some of the more recent studies concerning the use of CAP in dermatology and oncology while evaluating commonly used diagnostic and measurement techniques with the aim of elucidating major challenges in defining a unit dose of plasma medicine and offering a perspective for a future path towards overcoming this obstacle.

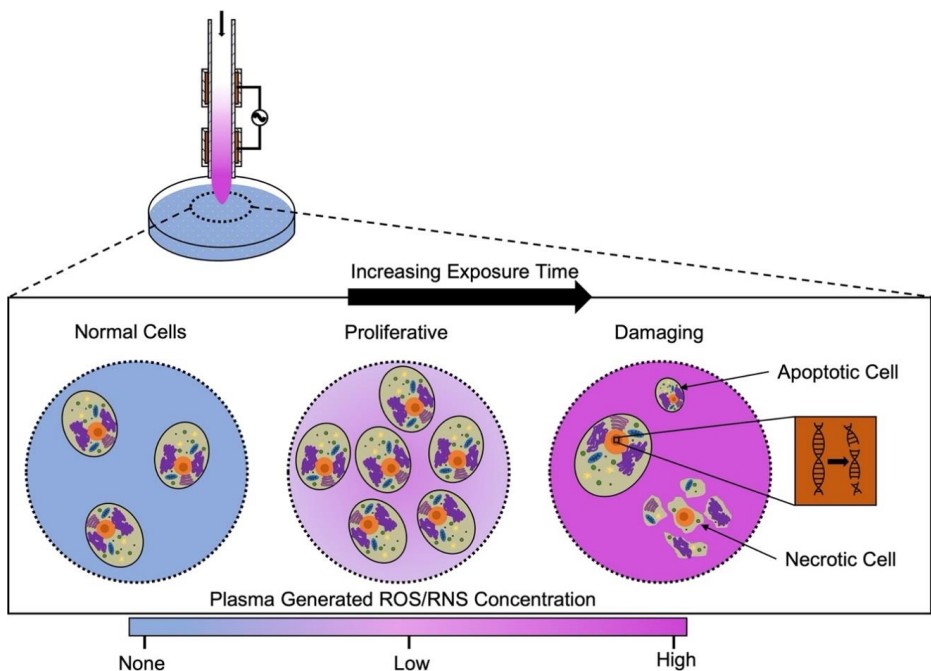

**Figure 4.** Hormesis effect of CAP. Illustration of the biphasic biological response to plasma treatment. Increasing the total exposure intensity causes the biological effects of CAP to change from proliferative to detrimental, pointing to a hormesis effect.

## 2. CAP and Dermatology

The most widely studied area of clinical application within plasma medicine is dermatology-related treatments. CAP treatments have been targeted for accelerated wound healing [53], disinfection [54], atopic dermatitis [55], wrinkle smoothing [56], scar treatment [57], and more [15]. There are a number of potential mechanisms of action for each of these applications. It has been hypothesized that the electromagnetic fields present in cold plasma devices may stimulate angiogenesis in the targeted area, which could be important for a number of dermatology-related concerns [14]. Mechanistic in vitro studies have suggested that exposure to CAP results in increased motility in both keratinocytes and fibroblasts due to altered regulation of Cx43, a gap junction protein that inhibits cell migration [58,59]. Evidence of increased cell proliferation and antibacterial effects of CAP treatments are also likely to play important roles in the efficacy of these treatments [9,60].

A selection of recently published studies, along with the types of data recorded and their methodology, are summarized below in Table 1. The variability of the experimental setup, treatment targets, types of data recorded, and methodology used for data acquisition between different studies is evident. These studies use a variety of plasma sources for their experiments, including commercially available, modified, and custom-designed plasma sources. Although variety in plasma sources will eventually help in distinguishing specific designs for specific applications, it does make the correlation of collected data across different studies very difficult. The recorded data can generally be broken into three categories: (1) input parameters and electrical characterization; (2) chemistry of treatment; and (3) biological response. Input parameters and electrical characterization include the power input to the device, plasma discharge power, excitation frequency, carrier gas composition and flow rate, and electrical characteristics of the plasma source. In general, the majority of these parameters can be measured accurately; however, plasma discharge power (power transfer efficiency of the plasma source, often dependent on excitation frequency) and electrical characterization of the plasma sources is more difficult. Regarding the chemistry of treatment, various spectrometric methods can be used to precisely measure the occurrence of certain species in the gas phase, including optical emission spectroscopy, mass spectrometry, IR spectroscopy, Raman spectroscopy, etc. These

methods lack the time resolution to detect important, highly reactive, short-lived species and lack the specificity for some others. For the in-liquid and intracellular measurement of chemistry, various colorimetric assays and spectrophotometric methods are utilized. However, these methods can often only detect one compound at a certain time point and lack the ability to provide real-time or time-resolved measurements. Although a wide range of very useful assays exist for the measurement of various physiological and cellular responses for in vitro applications, histology, biopsy, and blood samples are the only means of detecting such changes in vivo. Therefore, real-time and time-resolved measurements are often not possible for in vivo studies. Other physically visible physiological changes often can be recorded using various imaging techniques. Genomics and metabolomics are great methods of detecting physiological changes taking place at the cellular and tissue level post-CAP treatment. Such techniques allow us to measure gene expression, concentrations of certain proteins, and markers at specific time points before and after CAP treatment. Although such tools are commonly utilized for experimental purposes, they require significant sample processing and post-processing of data, making real-time measurements impossible. Additionally, performing all the various electrical, chemical, and biological characterization and measurements simultaneously is not feasible. Identification of the most impactful parameters in each of these realms could help reduce the number of measurements required. The combination of these measurement limitations elucidates the need for predictive modeling of CAP treatment outcomes employed in clinical settings.

**Table 1.** List of methods used and types of data produced in recent dermatology-related plasma-medicine studies.

| Plasma Source | Target | Recorded Data | Methods/Techniques |
|---|---|---|---|
| Cold atmospheric microwave plasma (CAMP) [61] | Open wound (canine) Immortalized human keratinocytes (HaCaT) Canine Progenitor epidermal keratinocytes (CPEK) | Input parameters: intensity of electric fields, frequency, ambient environment, treatment distance, etc. | Bio Stimulation Microwave Plasma V1.0, He plasma |
| | | Direct vs. indirect treatment | CAMP vs. activated media |
| | | Cell viability | CellTiter 96 Aqueous One Solution Assay Kit |
| | | Cell migration parameters | Scratch assay, Transwell migration assay |
| | | Wound area | Imaging |
| | | Gene/protein expression library | RNA-Seq, qRT-PCR, Blotting |
| DBD [62] | Thymoquinone treated with CAP was used to treat mouse models of wounds (in vivo) | Input parameters: intensity of electric fields, frequency, ambient environment, treatment distance, etc. | Suzhou Opus Plasma Technology Co. |
| | | Gene/protein expression | ELISA assays, Flow cytometry |
| | | Wound area | Imaging |
| | | Wound structure | Histology, Transmission electron microscopy (TEM) |
| Jet-DBD and planar DBD reactor [28] | Water treated with plasma under different conditions: gas composition, power | Input parameters: intensity of electric fields, frequency, ambient environment, treatment distance, etc. | In-house/modified plasma sources, He + $O_2$, $N_2$, or Air |
| | | Chemistry: $NO_2^-$, $H_2O_2$ | Griess assay, Spectrophotometric methods |

**Table 1.** *Cont.*

| Plasma Source | Target | Recorded Data | Methods/Techniques |
|---|---|---|---|
| RF-plasma jet [63] | *S. aureus*-infected wounds on Wistar rats with induced diabetes | Input parameters: intensity of electric fields, frequency, ambient environment, treatment distance, etc. | In-house/modified plasma sources, He plasma |
| | | Wound structure | Histology |
| | | Blood glucose level | Blood test |
| | | Epithelialization, inflammation, collagenization, vascularization | Histology |
| DBD- PlasmDerm FLEX9060 [64] | Prospective controlled cohort clinical trial-chronic leg ulcers | Input parameters: intensity of electric fields, frequency, ambient environment, treatment distance, etc. | In-house/modified plasma sources, He + $O_2$, $N_2$, or Air |
| | | Environmental parameters | Temperature, humidity |
| | | Patient data/medical history | Following patient confidentiality requirements |
| | | Wound area | Imaging |
| | | Capillary blood flow, tissue oxygen saturation, postcapillary venous filling, and microcirculation | Combined laser Doppler and photospectrometry |

As mentioned, due to the ease of application, dermatology has been the leading biomedical field in adopting CAP treatments. Promising studies in wound healing and sterilization have enabled early clinical application of plasma technologies. Brief descriptions of the current state of CAP treatments for wound healing and sterilization follow.

*2.1. Wound Healing*

Several mechanisms have been theorized as contributing agents to the efficacy of CAP treatment in accelerating wound healing. As mentioned, the electromagnetic field resulting from plasma generation could stimulate angiogenesis. Increased blood flow to the targeted region could partially explain improved wound healing. It has also been alluded that the CAP treatment creates a localized pH change that may contribute to wound healing [14]. Additionally, it is apparent that the antiseptic effects of plasma and wound sterilization play a role in faster wound healing [65].

Neutrophils are an especially important contributor to the wound healing process [66]. Neutrophils are the first cells to infiltrate the inflammatory site and are necessary for eliminating pathogens from a wound. An imperative function of neutrophils is the release of neutrophil extracellular traps (NETs), which are extracellular chromatin structures that are extruded by activated neutrophils to entrap and kill pathogens [67]. CAP treatments were shown to induce the formation of NETs in neutrophil cultures [67], which is expected as NET formation is known to be triggered via ROS signaling [67]. This pro-inflammatory response also encourages increased neutrophil migration to the site of inflammation. Furthermore, pro-inflammatory cytokines have been induced upon CAP treatment [66,68]. It is thought that these effects are a direct result of $H_2O_2$ generation from the plasma treatment. More specifically, it is possible that plasma-specific $H_2O_2$ enters into the neutrophil membranes, thus inciting the oxidation of redox probes and leading to respiratory burst [66]. A secondary theory for the effect of CAP on neutrophils is that the plasma-specific $H_2O_2$ interacts with myeloperoxidase, an enzyme abundant in neutrophils, and the two act synergistically to generate various other reactive species [66]. Although CAP treatments significantly affected the production of NETs in neutrophils, it was not found to have effects on other major neutrophil functions such as phagocytosis or the uptake of external pathogens [66]. Although CAP treatments are thought to be overall beneficial to the wound healing process, there is potential for treatments to be detrimental, depending

on dose, as NET formation has been shown to exacerbate diabetic wounds [66,67]. In fact, a recent in vitro study using a microsecond-pulsed helium jet concluded that treatments with their device, lasting longer than 60 s, were cytotoxic, reduced keratinocyte migration, and upregulated the expression of heat shock protein 27 (HSP27). This is while the same device used for treatments shorter than 60 s had no observed negative effects on keratinocytes, did not induce any change in mitochondria, and improved keratinocyte migration, leading to accelerated wound closure [69].

There have been several clinical studies surrounding the use of CAP treatments for wound healing. A multi-center controlled clinical trial investigating the efficacy of cold plasma therapy on chronic wounds revealed that patients treated with CAP experience significantly accelerated wound closure, while their need for antibiotic treatment was significantly reduced in comparison to those treated with standard wound therapy [70]. A group of researchers working on CAP treatment of diabetic wounds demonstrated an increase in growth factor (FGF-2 and VEGF-A), cytokine (IL-1$\alpha$ and IL-8), and TNF$\alpha$ levels in wound exudate samples within a prospective, randomized, patient-blinded, clinical trial [71].

### 2.2. Sterilization

Skin, as the body's first line of defense, plays an important role in passive immunity. However, several conditions can interfere with this organ's ability to keep infections out. For example, infection of chronic wounds and diabetic ulcers not only inhibit the wound healing process, but they can grow into the tissue and cause severe damage, resulting in sepsis, a need for amputation, or tissue removal. Another example is the heightened susceptibility of atopic dermatitis lesions to various infections. CAP has been demonstrated as a useful tool in clearing skin infections and, as a result, improving the wound healing process [53]. Furthermore, CAP treatments could potentially be used to treat viral skin conditions such as HPV warts [72,73].

The efficacy of cold plasma in inactivation of pathogens has been well established. In fact, the first application of cold plasma within the biomedical field was disinfection. Various studies have demonstrated that CAP treatment can inactivate a wide range of pathogens, including bacteria [9,74], fungi [75], and viruses [76]. With the rising occurrence of antibiotic-resistant bacterial strains, the appeal of a new anti-microbial approach such as CAP treatment is apparent. Owing to the nature of plasma, identifying the key constituents (e.g., reactive species, charged particles, free radicals, UV radiation, and free electrons) as the main effectors is challenging. That is because many cold plasma constituents, for example, UV radiation, are well known to be detrimental to pathogens. However, various studies have attributed reactive species as the major effector, noting that no single plasma constituent can account for the overall effect of CAP treatment on various microbial cultures [77–81].

The varied susceptibility of different microorganisms and the high variability of plasma discharge chemistry based on input parameters, as well as the gas medium used for plasma generation, further complicate this inquiry. For example, Gram-positive and Gram-negative bacteria exhibit different sensitivity levels to CAP treatments; some studies have claimed that a thicker cell wall results in less sensitivity to cold plasma treatment [82], whereas others have reported Gram-positive bacteria to be more susceptible than Gram-negative bacteria [83]. Considering the role of RONS in CAP treatments, various mechanisms of action can be theorized for the inactivation of bacteria, including cell wall oxidation and damage, metabolic changes, and genetic damage. A study of inactivation mechanisms involving several Gram-positive and -negative bacterial strains suggested irreversible poration in the cell wall of Gram-negative bacteria, whereas the reported mechanism for Gram-positive bacteria involved oxidation of cell wall and shrinkage [84]. An investigation into the subcellular mechanisms of inactivation of yeast yielded that, although damage to the membrane by hydroxyl radical ($^\bullet$OH) was likely the primary mechanism of inactivation, metabolic changes due to increased intracellular levels of singlet oxygen ($^1O_2$),

intracellular redox/ion imbalance, increased intracellular levels of hydrogen peroxide ($H_2O_2$), and DNA fragmentation all played a role in inactivation of yeast [85].

Cold plasma treatments have been shown to be effective against various antibiotic-resistant bacteria, even in cases for which the culture has grown to form bacterial biofilms [86,87]. A recent study evaluated the effects of CAP treatment on the most common multidrug-resistant bacterium, methicillin-resistant staphylococcus aureus (MRSA) [88]. This study used an APPJ with air as the carrier gas and was able to demonstrate that MRSA and non-drug-resistant strains of *S. aureus* had similar susceptibility to CAP treatment. Furthermore, they reported no development of resistance to CAP after 10 generations of exposure. The antiviral properties of CAP treatments are also of great interest. For example, various groups have demonstrated the efficacy of CAP treatment against SARS-CoV-2 [89,90]. The antiseptic properties of CAP have a wide range of applications within medicine and beyond, including the agriculture, food, and packaging industries.

### 3. CAP and Oncology

Cancer treatment is another major area of research within the field of plasma medicine. The majority of these studies have been conducted using cell lines or in vitro treatment methods. Cell lines for hepatomas [91], leukemia [92], brain tumors [93], and several other cancers, including colorectal [94], breast [95], lung [96], cervical [97], and head and neck [97] cancer is all strongly affected by cold plasma treatments. The majority of these cancer cell lines constitutively produce low levels of hydrogen peroxide in the nanomolar range [25]. However, supraphysiological levels of hydrogen peroxide introduced by CAP treatments are able to decrease proliferation and even induce apoptosis in the cancer cell lines via a caspase-dependent pathway [12,25]. Additionally, it has been hypothesized that the higher levels of basal $H_2O_2$ within cancer cells are responsible for the ability of cold plasma to selectively induce apoptosis in cancer cells over healthy host cells [98]. An alternative hypothesis for the susceptibility of cancer cells to CAP over healthy cells points to the reduced level of cholesterol in cancer cell membranes, which makes it easier to form pores in the cell membrane. Thus, it is easier to introduce the chemical effects of the CAP treatments [99]. A few studies have suggested that autophagy and Notch signaling may be affected during CAP exposure [98,100].

Although CAP treatments targeted for anti-cancer therapies have largely been studied for surface or skin-deep tumors, some models have shown that the chemical effects generated by plasma treatments have the ability to spread and disperse throughout deep tissue masses, penetrating phospholipid membranes [99]. In fact, multiple studies have demonstrated the ability of CAP to inhibit the growth of subcutaneous xenograft tumors and melanoma in mice via surface skin treatments [101–103]. Additionally, PAM treatments targeted against internal tumors that may be difficult to treat with topical approaches are under investigation [104]. The idea of using PAM over direct treatments has gained further traction due to the fact that PAM is less damaging for healthy cells, resulting in fewer morphological changes within the treated cells [105]. Moreover, a recent study investigated the biosafety profile of PAM injections into the bone marrow of rabbits, demonstrating no mortality or loss of mobility during the recording period [106].

An additional effect of PAM on immune cells that is of interest during cancer treatment is that macrophages are induced to release the inflammatory cytokine TNFα. This particular cytokine is important because it has been shown to block cancer cell growth in in vitro co-culture experiments with macrophages and glioblastoma cells. Aquaporin 8 (AQP8), a protein channel necessary for the transportation of water across the cell membrane [107], may also play an important role in the anti-cancer effects of CAP treatments. Yet another theory for the effectiveness of CAP treatments against cancer cells states that plasma treatments, either direct or indirect, have the ability to increase the production of damage-associated molecular patterns (DAMPs). DAMPs are molecules that are released from cells that have been damaged or may be dying as a result of trauma or inflammation, often induced by an invading pathogen [108]. These molecules are crucial for the immune

system to recognize active cancer cells and initiate a more robust defensive response. Specifically, PAM treatments have been demonstrated to trigger cell death and increase DAMPs in both melanoma and pancreatic cancer cells in vitro [98]. On the other hand, it is speculated that the tumor microenvironment may play a role in blocking CAP treatments as it has previously been shown to interfere with ionizing radiation and chemotherapeutic treatments [98]. Table 2 summarizes some of the recent work in this field. The overall advantages and limitations of the types of methods mentioned in this table are discussed with respect to Table 1.

**Table 2.** List of methods used and types of data produced in recent oncology-related plasma medicine studies.

| Plasma Source | Target | Recorded Data | Methods/Techniques |
|---|---|---|---|
| DBD [109] | Cultures of esophageal cancer cell lines EC9706 and ECa109 in DMEM medium with 10% FBS | Input parameters: intensity of electric fields, frequency, ambient environment, treatment distance, etc. | In-house plasma sources, Air plasma |
| | | Cell viability | MTT assay, Apoptosis assay |
| | | Chemistry in medium: $NO_2^-$, $NO_3^-$, $H_2O_2$ | Spectrophotometric techniques and assays |
| | | Levels of glutathione, intracellular ROS | Hydrogen peroxide assay, flow cytometry |
| APPJ [110] | Human squamous cell carcinoma cell line A431 and skin malignant melanoma cell line A375 in RPMI 1640 medium with 10% FBS, 1% glutamine, and 1% penicillin + streptomycin | Input parameters: intensity of electric fields, frequency, ambient environment, treatment distance, etc. | kINPen argon plasma, dose of indirubin added |
| | | Cell viability and metabolic activity | Alamar blue assay, sytox blue assay + flow cytometry |
| | | Cell migration | Scratch assay |
| APPJ [37] | DI water used to prepare Pluronic hydrogels intended for intratumoral injection post-plasma treatment | Input parameters: intensity of electric fields, frequency, ambient environment, treatment distance, etc. | In-house plasma sources, air plasma |
| | | Gas phase chemistry | Optical emission spectroscopy |
| | | Intracellular ROS and RNS | Stained with fluorescent dyes + flowcytometry |
| | | Cell viability, tumor size, and immunological analysis | IVIS imaging and various antibodies + flowcytometry |
| APPJ [111] | Various human cancer cell lines treated directly or via PAM (in vitro) Topical treatment of melanoma tumors on mice (in vivo) | Input parameters: intensity of electric fields, frequency, ambient environment, treatment distance, etc. | MediPL plasma torch system, Argon plasma |
| | | Gene/protein expression | qRT-PCR, Western blot |
| | | In vivo apoptosis/protein expression | Immunohistochemical studies |

There is now a consensus in the field that cold plasma therapy will likely be a powerful tool for the treatment of a variety of cancers when combined with existing therapies. For example, a major problem with surgical removal of solid tumors is the high probability of residual cancer cells remaining on the tumor bed, leading to a high reoccurrence rate and, as a result, high mortality. Canady Helios Cold Plasma (CHCP) is a promising novel cold plasma source for the treatment of solid tumors. This is the first cold plasma device to successfully undergo an FDA-approved phase-I clinical trial for the treatment of advanced solid tumors. This clinical trial took place between March 2020 and April 2021 and was conducted with 20 stage-IV patients (patients with metastasis). These studies showed an overall survival rate of 24% at 31 months. R0 resection (no microscopic residual tumor in surgical margin) was achieved in 12 patients; however, microscopic cancer cells were

identified at the margin site in five patients. These five patients had an overall survival rate of 40% at 28 months. However, there were no signs of recurrence (no regrowth of tumor at the primary tumor site) during the follow-up period, which signals the efficacy of CHCP in inactivation of any undetectable microscopic tumor cells remaining in the surgical boundaries [33].

## 4. Intracellular Targets of CAP

The effects of CAP on the immune system and specific signaling pathways impacted during CAP treatments are increasingly being investigated. Figure 5 depicts a broad overview of the effects of CAP on targeted host cells. As discussed above, melanoma and pancreatic cancer cells have shown an increase in DAMP production, which may trigger apoptosis in these cells [98]. Additionally, cancer cells, in general, appear to have increased sensitivity to CAP-specific ROS [98,99]. Lastly, an important role for aquaporins during CAP treatment has been proposed in cancer cells [12]. Epithelial cells have displayed an increase in proliferation and promotion of angiogenesis following CAP treatment [59,60]. Additionally, the impact on immune cells such as neutrophils and macrophages has been studied. Neutrophils have increased NET production and potential interactions of CAP-specific $H_2O_2$ with myeloperoxidase, resulting in an accumulation of reactive species [66]. Macrophages are preferentially shunted towards an M1, pro-inflammatory phenotype following CAP treatment, initiating various pro-inflammatory signaling cascades [10,60]. Finally, there has been very minimal study of the impact of cold atmospheric plasma on T and B cells. However, a study investigating T cell leukemias did find an increase in T cell apoptosis following CAP treatment [112].

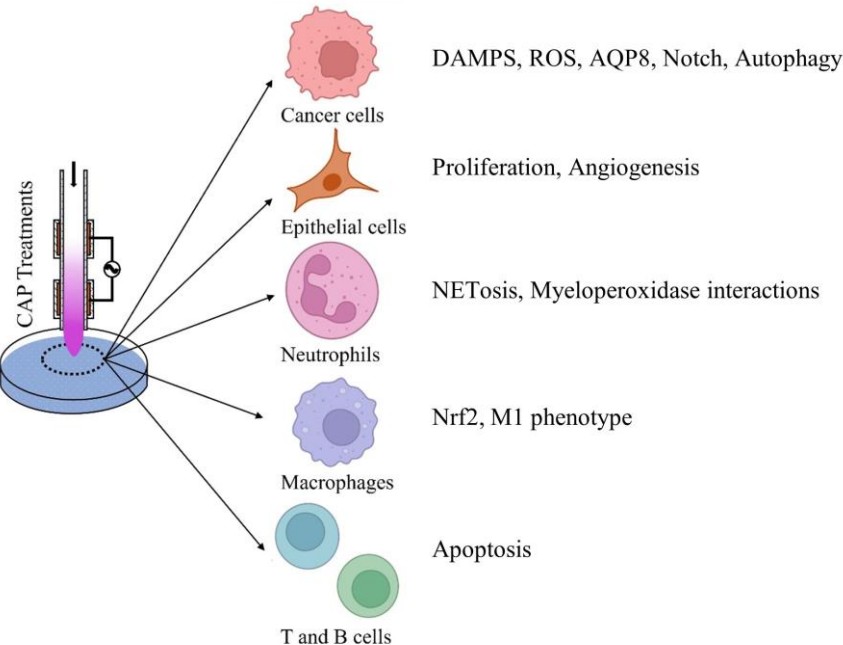

**Figure 5.** Effects of CAP on host cells. Brief description of potential effects of CAP on different host cell groups including cancer cells, epithelial cells, neutrophils, macrophages, T and B cells. Potential effects were extracted from the recent literature.

Figure 6 gives an overview of some of the signaling pathways and intracellular targets of CAP and alludes to potential molecular mechanisms behind the effects of the treatments. As discussed, CAP induces NETosis and interacts with myeloperoxidase in neutrophils. Macrophages are another well-defined component of host defense during inflammatory conditions. The effects of CAP on macrophages have been studied in the context of both dermatology and oncology CAP treatments [10,113]. One immunogenic response that

has been specifically investigated during dermatology-associated CAP treatments is the induction of the Nrf2 pathway [10]. Activation of the Nrf2 pathway has been known to regulate the infiltration of additional immune cells to the site of inflammation, and regulation of angiogenesis, which is crucial for would healing processes and recovery in other dermatology-related issues [114]. The downstream effects of Nrf2 activation can also contribute to other signaling cascades via the production of both pro- and anti-inflammatory cytokines [10]. Studies have also indicated that CAP treatments can polarize macrophages towards an M1, pro-inflammatory phenotype, thus inducing the release of cytokines including TNFα, IL-12, and IL-1β, as well as increasing levels of iNOS [113]. As discussed, TNFα is a cytokine of particular interest during cancer treatments. These combined soluble factors improve the anti-tumorigenic immune responses in several in vitro settings [113].

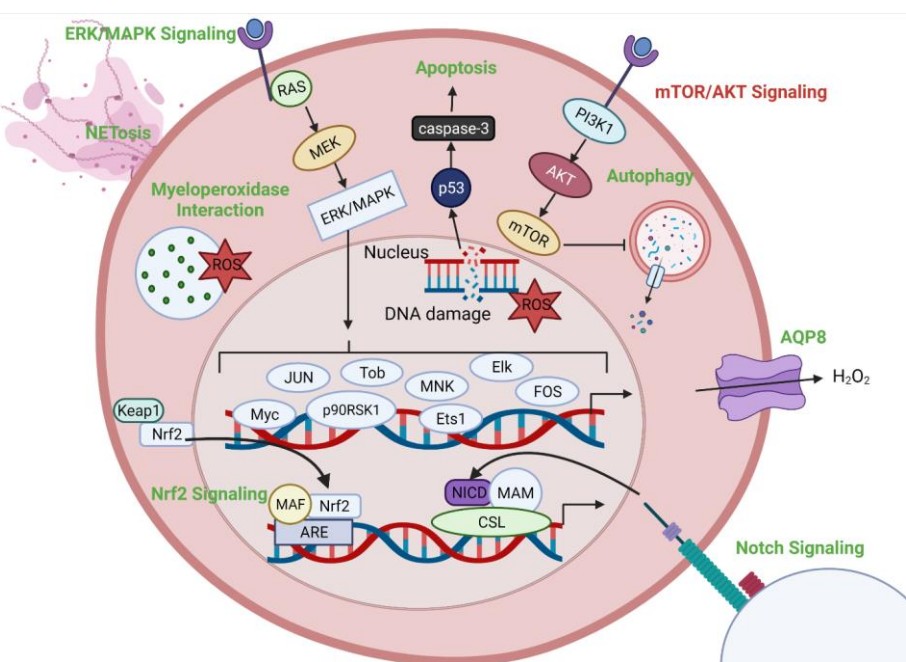

**Figure 6.** Potential cellular targets of CAP treatment. Intracellular pathways and effects are depicted. Pathways upregulated following CAP treatment are displayed in green. Pathways downregulated following CAP treatment are displayed in red.

A recent study describing potential mechanisms for cancer cell resistance to CAP treatments demonstrated that another important intracellular effect following exposure to CAP was an increase in Notch signaling. This was specifically demonstrated in prostate progenitor cells using an in vitro primary cancerous prostate basal epithelial cell culture model [100]. The Notch pathway is important for cell death, differentiation, and proliferation and also plays a role in tumorigenesis [115]. Furthermore, Notch signaling has been identified as a regulator of oxidative stress [100]. Interestingly, this pathway has also been studied in the context of radiation therapy treatments and has been identified to present similar resistance mechanisms under those treatment conditions [116]. Therefore, combination therapies, including Notch inhibitors, have been proposed to increase the efficacy of CAP anti-cancer treatments [100].

The protein kinase B (AKT), mammalian target of rapamycin (mTOR), and extracellular signal-receptor kinase/mitogen-activated protein kinase (ERK/MAPK) pathways have also been investigated during CAP treatments. AKT is a kinase that is known to regulate various cellular functions, including metabolism, proliferation, and survival [117]. However, AKT also plays a role in tumor development under oncogenic conditions [117,118]. Both in vitro and in vivo models of head and neck cancer revealed that CAP treatments increased expression of mitochondrial E3 ubiquitin protein ligase 1, which inhibited AKT levels

corresponding to an anti-cancer effect [118]. mTOR signaling is downstream of the AKT pathway. Therefore, it is unsurprising that it is also notably affected during CAP treatments. However, it is important to note that the inhibition of mTOR via CAP treatments may result in the activation of autophagy pathways. Autophagy is a highly conserved, catabolic, self-degrative process used to balance energy sources. This pathway can be induced by various cellular stressors and is especially important during conditions of nutrient deficiency [119]. Significant increases in autophagy following plasma treatment have been shown to lead to Type II or autophagic cell death in targeted endometrial cancer cell lines [120]. Finally, the ERK/MAPK pathway contains a series of protein kinase cascades and regulates genes involved in cell proliferation, differentiation, and development via a variety of transcription factors [121]. Studies centered around the MAPK-p53 axis revealed a CAP dose-dependent biphasic response in the activity of the ERK/MAPK pathway. Specifically, the pathway was activated at low treatment intensities, which was hypothesized to reflect the importance of short-lived reactive species generated in the plasma [112,121].

As previously mentioned, AQP8 may also play an important role in the immunogenic effects of CAP treatments. Evidence of this has been shown via decreased effectiveness of anti-cancer plasma treatment upon inhibition of AQP8 [12]. It is possible that inhibition of AQP8 decreases the permeability of the cells, thus decreasing the influx of CAP-generated reactive species. It has also been suggested that the variation in aquaporin expression among different cell lines could explain inconsistencies in the effects of CAP treatments [25]. Finally, the ability of CAP treatments to induce apoptosis in targeted cells has been demonstrated via a caspase-dependent mechanism [122]. Interestingly, a study using Jurkat cells, which are an immortalized T cell leukemia cell line, demonstrated increases in both p53 and caspase-8 following treatment with CAP. Furthermore, increases in levels of intracellular ROS and resulting DNA damage were also observed. As p53 and caspase-8 are elements of independent apoptotic pathways, the combination of these factors resulted in an increase in apoptosis in the CAP-treated samples [122].

Cold plasma–treated cells experience a high level of oxidative stress due to the exogenous ROS introduced during treatment. In fact, the change in the redox balance of cells as a result of CAP treatment has been reported in several studies [123–125]. One study used glutathione (GSH) as a marker for intracellular and extracellular oxidation levels due to CAP treatment. They observed a continuous oxidation of extracellular GSH, suggesting the emergence of a stable exogenous oxidant into the media via CAP treatment [125] Although they reported mitochondrial oxidation (high GSH levels in mitochondria) and membrane depolarization post-CAP treatment, there was no direct correlation between oxidative stress and intracellular GSH levels. Several such studies have drawn conclusions of a plasma-dose-dependent redox imbalance in cells leading to apoptosis at higher treatment intensities. This has instigated an innovative approach within the field to treatment normalization and dose definition. Several groups are investigating the measurement of biomarkers and biochemical indicators as a means of defining CAP treatment dosage. Extracellular glutathione oxidation levels are one example of such biomarkers [126]. Another example used plasma input parameters such as carrier gas flow rate, voltage, and time to predict cell viability based on previous data [49].

Although such predictive models based on limited physical parameters can be reliable for the specific experiment, the model would have to be reconstructed and verified for every plasma source and every treatment target. A more generally applicable predictive model introduces a biochemical parameter, equivalent total oxidation potential (ETOP) for a unit dose of PAM [127] The parameter, ETOP, is introduced as a function of three components. The first component is the oxidative potential of CAP produced RONS, the second component is defined as the oxidative potential of other plasma components, and the third component is defined as any synergistic effect that may arise from the combination of the first two components. ETOP is meant to correlate RONS levels to the biological outcomes of specific target cell types, via modeling of dose–response relationship and, therefore, is proposed to be accurate irrespective of intermediary chemistry or plasma

source. Fundamentally, CAP chemistry and biological response to RONS treatment are individually characterized and correlated via a dose–response model [127].

## 5. Conclusions and Future Perspectives

The widespread use of CAP across multiple disciplines within the medical field is unsurprising given the wide range of impact of ROS and RNS in many living organisms and the indications of a biphasic dose response providing the opportunity for a variety of effects from the same treatment. Although dermatology and oncology are the main foci of this article and encompass the majority of recent experimental work within plasma medicine, experts in neuroscience, dentistry, and ophthalmology are investigating potential applications of plasma medicine in their respective fields. Despite significant progress in the field of plasma medicine, there are still fundamental challenges that must be overcome. One major challenge is establishing a definition for the unit dose of CAP treatments that is easily measurable and controllable.

First and foremost, the role of RONS as the primary bioactive effector of CAP treatments is well accepted within the field. Therefore, we can claim that the main biological impacts of such treatments are mediated by biochemical interactions, of which we have limited understanding. Thus far, it has been established that CAP treatments exhibit a hormesis effect, meaning that lower exposures induce proliferation, whereas higher exposures are detrimental to mammalian cells. Additionally, there is a variability of biological responses to identical CAP treatment across different treatment targets, meaning that the complex and temporally changing chemical composition of CAP discharge could exhibit different interactions within various physiological systems. Secondly, the chemistry of CAP discharge is highly dependent on various input parameters (e.g., power, frequency, ambient conditions, etc.), discharge regime (e.g., APPJ, DBD, SBD, etc.), and device configuration (e.g., geometry, materials, etc.). Lastly, there is a general need for specialized plasma diagnostic equipment. For instance, plasma discharge chemistry is currently measured using various spectroscopic techniques such as optical emission spectrometry (OES). Such techniques are widely used to measure more abundant and stable chemical species (e.g., hydrogen peroxide, nitrates, nitrites, various atomic and molecular transitions, etc.); however, they can only detect gas-phase chemistry, they lack the spatial resolution for precise detection near the plasma/target interface, and at temporal resolutions necessary to detect highly reactive (short-lived) species real-time or near real-time measurement is not possible (requires extensive post-processing). Other techniques, such as mass spectrometry and electrical probing, are not suitable for complex chemistries, lack sensitivity, or interact and alter the discharge. PAM and cellular chemistry changes are detected using a variety of colorimetric assays and spectrophotometric techniques or electrochemical probes, which either lack the temporal resolution necessary or are highly specific to individual compounds and lack the ability to detect the full range of plasma chemistries. Furthermore, real-time detection of plasma chemistry in tissue is not possible, and current techniques have very little to no temporal resolution. The complexity and variability of CAP treatments, combined with the lack of efficient diagnostic techniques, create a need for the use of predictive modeling to determine and measure a unit dose of CAP treatment. The utilization of "state-of-the-art" measurement techniques such as time-resolved attenuated total reflectance–Fourier transform infrared spectroscopy (Time-resolved ATR-FTIR) [128] and further development of such diagnostic techniques will prove essential to the advancement of the field of plasma medicine.

The utility of CAP in biomedicine is undergoing rigorous testing and demonstrating significant potential. Cold plasma treatments have been shown to possess selectivity towards cancer cells and have proved to be a reliable source for the treatment of various skin maladies and infections. The promising efficacy of cold plasma in these fields has generated an abundance of interest and a vast collection of data on a variety of cancer types, cell types, bacterial strains, and other biological test samples. Researchers working on different cancer types, for example, have published data linking the selectivity of CAP

treatment to several diverse mechanisms of action with each target. These mechanisms range from p53-linked apoptosis activation and cell cycle arrest to DNA damage caused by an elevated intracellular $H_2O_2$ concentration and functional changes to membrane receptors. Yet, the majority of alluded mechanisms focus on plasma-generated chemistry (e.g., ROS and RNS).

The overlap between redox biology and cold plasma chemistry makes it quite clear that plasma-generated ROS and RNS are the main bioactive constituents of CAP treatments; however, it could be naïve to ignore the probable physiological effects and side effects of other plasma constituents, such as UV radiation, free electrons, radicals, and the electromagnetic field. Combining this fact with the rising popularity of clinical CAP treatments around the world, especially the promising progress of the first biomedical cold plasma source via the FDA clearance process in the US, the need for a much more in-depth understanding of CAP treatments is abundantly clear. Multiple groups are currently investigating the chemical, electrical, electrochemical, optical, and mechanical characteristics of various cold plasma sources. Others are focusing on the biochemical interactions resulting from CAP treatments. Expansion of the currently available data on the cellular response, the systemic effects of CAP treatment, and its correlation with the plasma source characterizations could help overcome the major obstacles to widespread clinical adoption of plasma medicine, which are treatment normalization and dose definition.

The range of potential plasma sources, operating parameters, environmental factors, biomedical targets, and measurable parameters would appear to make defining a unit dose and developing a form of predictive control of outcomes a nearly intractable problem. We propose that advances in data science, particularly artificial intelligence and machine learning (AI/ML) techniques, coupled with data curation and protocols for sharing amongst the plasma-medicine community, can accelerate the progress of the field towards greater societal impact. AI/ML has demonstrated considerable promise in materials science [129–131] and nuclear fusion [132–134], among many other fields. The commonalities among these applications are as follows: (1) a vast information database of many independent investigators (e.g., condensed matter science) or (2) a large database from a singular experiment with variable parameters (e.g., DIII-D). The plasma medicine community cannot likely achieve the latter without sustained funding of an agreed-upon experimental configuration, but the former is possible with an agreed-upon reporting method among different research groups. Examples include the Plasma-MDS metadata approach [135], which could be included as supplementary data in every plasma medicine article, allowing members of the community to download and integrate it into AI/ML models.

Figure 7 depicts an arbitrary plasma medicine system, as well as a non-comprehensive list of recordable or measurable parameters. An additional challenge to this large-scale data collection is that not every research group possesses the same capabilities, although a number are broadly applicable. For instance, all research groups should be capable of measuring and reporting operational parameters such as the environmental conditions or controlled inputs such as gas sources, as well as the device type and associated design parameters. The measurement of applied voltage or plasma input power is similarly relatively straightforward in well-resourced laboratories. The measurement of gas-phase chemistry becomes more complicated, in which passive optical emission spectroscopy is relatively accessible with modular spectrometers, but more advanced time-resolved techniques with greater sensitivity, such as laser-induced fluorescence or those using intensified detectors for passive spectroscopy, are not necessarily available. Next, one of the greatest diagnostic challenges in the field is the characterization of the gas–tissue (liquid) interface due to the short temporal and spatial scales of radical species reactions. However, those reactions may be critical to characterizing plasma medical effects and, thus, need to be better understood. The plasma-medicine community should attempt to adapt techniques from other fields, such as materials science, in which thin film chemistry is commonly measured. One such technique, time-resolved infrared spectroscopy (TRIR), holds considerable promise [136,137] but will require substantial control of target thickness,

such as gelatins, hydrogels, or tissue, for accurate measurement given the desiccating effects of many plasma sources. The technique still has drawbacks associated with sensitivity to certain species (e.g., atomic species), complex instrument requirements, and spatial extent, requiring many experiments to characterize the evolution of reactive species as a function of depth into a target. Cell response measurements are well-established and are primarily a case of access to the requisite resources such as culture hoods, microscopy, flow cytometry, multi-omics techniques, etc. Additional opportunities arise when considering the integration of numerical simulation of plasma-induced chemistry and interaction with biological or abiotic targets or improvement in these numerical models with the proposed AI/ML approach. These are outside the scope of this review, but there are excellent reviews available for interested readers [138–140].

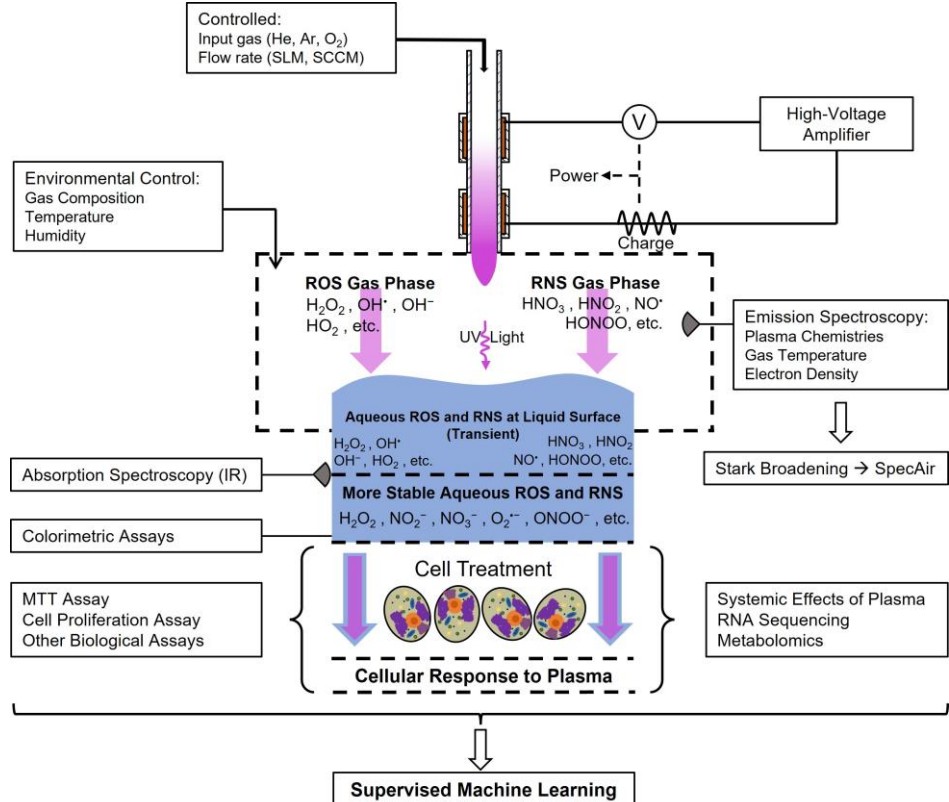

**Figure 7.** Future directions for plasma medicine. A simplified demonstration of the overall vision for the evolution of plasma medicine. Using multiple measurement techniques, AI/ML models may be able to identify key parameters from which the community can predict the biological effects of plasma treatments, ideally resulting in accelerated societal impacts. Arrows depict diffusion, reaction, or treatment (blue: treatment substrate, violet: plasma constituents).

The ideal goal of this approach is the identification of (1) controllable inputs, such as device design, applied voltage, or ambient environment, or (2) accessible real-time diagnostics, such as passive optical emission spectroscopy coupled with (3) the biological target or application to predict and control the biological response. It is also possible that such an approach would identify a different diagnostic, such as LIF or TRIR, as the most critical measurement. Although these are not currently available to apply in vivo, presently, knowledge of this need would accelerate development to make it possible.

**Author Contributions:** A.K. and M.J.N. contributed equally to the literature search, organization, and writing of the manuscript. S.D.K., S.G.B. and G.S.K. supervised the literature search, co-wrote the review, and revised the manuscript. All authors have read and agreed to the published version of the manuscript.

**Funding:** This study was partially supported by the National Institutes of Health under Awards NCATS TL1TR002016 to M.J.N., the NIBIB R21EB024693 to S.D.K., S.G.B., and G.S.K, and USDA-NIFA Hatch project # PEN04771 to GSK. The content is solely the responsibility of the authors and does not necessarily represent the official views of the National Institutes of Health. We would like to acknowledge The Pennsylvania State University Center for Biodevices Seed Grant Program, The Huck Institutes of the Life Sciences, and The Pennsylvania State University College of Engineering Diefenderfer Fellowship for their part in the financial support of AK during the draft process.

**Conflicts of Interest:** The authors declare no conflicts of interest.

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
