# Peer review of "Cold Atmospheric Plasma Medicine: Applications, Challenges, and Opportunities for Predictive Control"

_plasma, doi:10.3390/plasma7010014_

Round 1

Reviewer 1 Report

Comments and Suggestions for Authors

The submitted manuscript reviews the state of the art in the field of plasma medicine, covering both the technical background and current biomedical research. Dermatological and oncological applications are covered as well as effects on the immune system and intracellular signaling pathways. The selection of literature on which the review is based appears to be adequate and includes mainly recent and very recent publications. The authors have succeeded in structuring and classifying the cited papers and selectively summarizing the content. The conclusion is coherently derived from the previous chapters and proposes concrete measures to improve the prediction of treatment outcomes. The authors suggest that plasma medicine can benefit from interdisciplinary collaboration, data-driven approaches, advanced characterization techniques, and clinical trials to optimize and validate CAP treatments for various diseases and applications.

The manuscript is well structured, very well written and its content is relevant for the plasma medicine community. I recommend publishing it in Plasma after addressing two minor issues that do not require a further review:

Page 3, line 109: materials including polycarbonate…

I am not aware of any application of polycarbonate as a dielectric in a DBD application. Please provide an example.

Tables 1 and 2

The table captions seem to contain a typing error, I guess it should say List of the types of data…

Author Response

Thank you so much for your review of this work. We are grateful and appreciative of the time you have dedicated to reading this manuscript and providing thoughtful comments. After careful and thorough consideration of every comment, we reflected them to the best of our ability. We believe that these revisions have helped improve the manuscript and prepare it for publication. Please see below for a list of corresponding revisions:

1. The sentence was adjusted and polycarbonate was removed.  Although our group has used polycarbonate as a dielectric in various device configurations, we have not published these works.

2. Table captions have been adjusted.

You can find these changes in the resubmitted document highlighted in green.

Reviewer 2 Report

Comments and Suggestions for Authors

This manuscript provides a comprehensive overview of the research in plasma medicine field mentioning in particular studies in dermatology and oncology as well as mechanisms at the cell levels responsible for antimicrobial effects. The text encloses both fundamental research and ongoing clinical investigations, also providing current state-of-the-art. It also makes a retrospective of studies dealing with effects of direct and indirect effects of plasma treatments which makes the text a valuable and inclusive source of information. Thus, I found the manuscript extremely useful, informative and with well-balanced level of information related to plasma physics, bio-chemistry and bio-medicine. Thus, I fully support the publication in Plasma as it will found readers among experts from many different fields. I propose only minor changes to the text.

Lines 123-127: I would add the possible effect of ions in the explanation. They also have an important role in biosystems and at atmospheric pressure they have very short mean free paths before collision and recombination. Thus, the distance from the active plasma volume may also affect the treatment and reduce its effect.

Lines 300-303: It would be useful to provide a time-span used for bibliography search in order to have a complete information.  

Line 317: Mass spectrometry is an additional technique that can provide mostly qualitative insight to both neutral and ionic species formed in the plasma. It is maybe worth mentioning. 

Lines 695-707: I would not neglect the contribution of creation of more complex computer models for describing complex interaction between CAP and biological systems. 

Author Response

Thank you so much for your review of this work. We are grateful and appreciative of the time you have dedicated to reading this manuscript and providing thoughtful comments. After careful and thorough consideration of every comment, we reflected them to the best of our ability. We believe that these revisions have helped improve the manuscript and prepare it for publication. Please see below for a list of corresponding revisions:

1. This explanation was expanded to include recommended points.

2. The time span of the literature search was added in lines 48-49

3. As you mentioned mass spectrometry is a common technique that was missing from the list. Therefore it has been added in lines 316-317.

4. A few sentences discussing the importance of other computer models have been added.

You can find these changes in the resubmitted document highlighted in Turquoise.